# HIV-1 Proviral Genome Engineering with CRISPR-Cas9 for Mechanistic Studies

**DOI:** 10.3390/v16020287

**Published:** 2024-02-13

**Authors:** Usman Hyder, Ashutosh Shukla, Ashwini Challa, Iván D’Orso

**Affiliations:** Department of Microbiology, The University of Texas Southwestern Medical Center, Dallas, TX 75390, USA; usman.hyder@utsouthwestern.edu (U.H.); ashutosh.shukla@utsouthwestern.edu (A.S.); ashwini.challa@utsouthwestern.edu (A.C.)

**Keywords:** HIV-1, Tat, viruses, transcription, CRISPR, Cas9, genome engineering

## Abstract

HIV-1 latency remains a barrier to a functional cure because of the ability of virtually silent yet inducible proviruses within reservoir cells to transcriptionally reactivate upon cell stimulation. HIV-1 reactivation occurs through the sequential action of host transcription factors (TFs) during the “host phase” and the viral TF Tat during the “viral phase”, which together facilitate the positive feedback loop required for exponential transcription, replication, and pathogenesis. The sequential action of these TFs poses a challenge to precisely delineate the contributions of the host and viral phases of the transcriptional program to guide future mechanistic and therapeutic studies. To address this limitation, we devised a genome engineering approach to mutate *tat* and create a genetically matched pair of Jurkat T cell clones harboring HIV-1 at the same integration site with and without Tat expression. By comparing the transcriptional profile of both clones, the transition point between the host and viral phases was defined, providing a system that enables the temporal mechanistic interrogation of HIV-1 transcription prior to and after Tat synthesis. Importantly, this CRISPR method is broadly applicable to knockout individual viral proteins or genomic regulatory elements to delineate their contributions to various aspects of the viral life cycle and ultimately may facilitate therapeutic approaches in our race towards achieving a functional cure.

## 1. Introduction

Several molecular processes are essential components in the HIV-1 life cycle, from binding to target cell receptors, the reverse transcription of viral RNA, import of viral DNA into the nucleus, formation of a pre-integration complex, integration into host genomic DNA, transcription, RNA export, translation, and transport of proteins and genetic material into the cytoplasm for the assembly, budding, and maturation of the virion [1,2]. Transcription, the conversion of the integrated provirus to functional RNA units, is a critical regulatory process that relies on several host and viral transcription factors (TFs), including the HIV-1 Tat protein [3,4,5,6,7,8,9,10,11].

The HIV-1 transcriptional program is composed of three discrete phases (basal, host, and viral) distinguished by cell state and what TFs operate in each phase (Figure 1) [12]. During the basal phase in resting cells (CD4^+^ T and myeloid), the current model poses that the provirus is in a repressed chromatin state, which restricts HIV-1 expression and, thereby, Tat synthesis [13,14,15]. When resting cells are exposed to signals from the immune microenvironment, the host phase is initiated through signaling cascades that culminate in the transcription activation of the provirus via a plethora of mechanisms, including host TF activation through translocation from the cytoplasm into the nuclei (e.g., NF-κB and NFAT) and/or transcription and post-translational modifications (e.g., AP-1) [16,17,18,19]. The binding of these TFs to their cognate cis-elements at the provirus facilitates several steps in the transcriptional program, leading to latency reactivation through the action of RNA Polymerase II (Pol II) and co-activators, such as the P-TEFb kinase. Upon recruitment to the HIV-1 promoter [20,21,22,23], P-TEFb phosphorylates Pol II and factors that stabilize Pol II pausing [24,25,26,27,28], to facilitate Pol II pause release, which is as a key rate-limiting step for the early induction of HIV-1 transcription [27,29,30,31,32].

During the host phase, Tat is expressed, which promotes the transition to the viral phase for robust HIV-1 gene transcription (Figure 1). Upon Tat expression, a constant source of P-TEFb is recruited to the nascent trans-activation response (TAR) RNA to assemble Tat:P-TEFb:TAR ternary complexes, thereby facilitating sustained pause release and processive elongation. Together, this creates a positive feedback loop (Figure 1), where Tat continuously activates high levels of HIV-1 transcription, enabling robust viral replication and pathogenesis.

Progress in understanding HIV-1 transcription has been made by a large body of work that has used *tat* transfections in reporter assays to induce the viral promoter [7,8,33,34]. However, these studies have ignored the host phase of HIV-1 transcription thus rendering it impossible to determine whether factors operate prior to and/or after Tat synthesis, underscoring the need for genetic systems that allow the investigation of both host and viral phases in the transcription of integrated proviruses (Figure 1). To circumvent this limitation, we devised a proviral genome-editing approach with CRISPR-Cas9 to mutate the *tat* translation initiation codon (ATG) in order to eliminate expression of the Tat protein in immortalized CD4^+^ T Jurkat latency (J-Lat) cell models. In this method, cells were electroporated with a functional editing ribonucleoprotein (RNP) complex consisting of the Cas9 endonuclease along with a CRISPR RNA (crRNA)-transactivating crRNA (tracrRNA) duplex, which tethers Cas9 to the *tat* locus to create a double-strand break. After cleavage, the DNA is repaired using Homology-Directed Repair (HDR) with a donor template, resulting in a modified sequence.

Using this CRISPR approach, we created a genetically matched pair of T cell clones harboring HIV-1 at the same integration site, with and without Tat. In the wild-type clone, HIV-1 is regulated through the sequential action of host and viral phases, while in the Tat-minus clone, HIV-1 is regulated solely by the host phase (Figure 1). The molecular characterization of these T cell clones allowed us to distinguish between the host and viral phases of the HIV-1 transcriptional program. We propose that this system will allow future studies to examine the precise molecular characterization of HIV-1 transcription-related events prior to and after Tat synthesis, including chromatin accessibility, nucleosome positioning, histone modifications, and Pol II regulatory mechanisms such as recruitment, pausing, pause release, elongation, and termination. Notably, this system can also help validate whether Latency Reversing Agents (LRAs) [35,36] and Latency Promoting Agents (LPAs) [37,38,39] function by targeting the expected host or viral factors during the three distinct phases of the HIV-1 transcriptional program, thus guiding future therapeutic opportunities.

## 2. Methods

### 2.1. Cell Culture

J-Lat 10.6 (10.6) cells, and all clonal lines generated, were cultured in RPMI-1640 Medium (MilliporeSigma, Saint Louis, MO, USA, R8758) supplemented with 7% Fetal Bovine Serum (FBS) (MilliporeSigma, Saint Louis, MO, USA, F4135) and 1% Penicillin/Streptomycin (P/S) (Gibco, Grand Island, NY, USA, 15140163) at 37 °C with 5% CO_2_. After electroporation, RPMI-10%FBS-1%P/S was used to promote cell growth. Mycoplasma tests (Southern Biotech, Birmingham, AL, USA, 13100-01) were conducted frequently to ensure that cells were not contaminated throughout the completion of this study. The original 10.6 cell model was isolated by the infection of Jurkat CD4^+^ T cells with a full-length but replication-defective (Env-) HIV-1 (R7/3) genome [40]. Four days post-infection, GFP-negative cells were isolated and treated overnight with TNF-α. GFP+ cells were then sorted into single-cells and expanded to generate clones. Among these latently-infected clones was the parental J-Lat 10.6 cell model used in this paper. 

### 2.2. Cell Treatments

Human Tumor Necrosis Factor-α (TNF-α) (PeproTech, Cranbury, NJ, USA, 300-01A) was used at a final concentration of 10 ng/mL for 16 h (hours) with a vehicle control containing Phosphate-buffered saline (PBS) to build the cell line. Phorbol 12-myristate 13-acetate (PMA) (MilliporeSigma, Burlington, MA, USA, P8139) and Ionomycin (MilliporeSigma, Saint Louis, MO, USA, I0634) were used for treatments marked “PMA” (10 ng/mL of PMA and 0.1 µM of Ionomycin) with DMSO treatment as the corresponding control at the indicated time points in different experiments (16 h for western blots, 16 h for flow cytometry, and 0.5, 1, 4, 16 h for the RT-qPCR and ChIP-qPCR time courses).

### 2.3. Generation of 10.6 Tat^KO^ and Ctrl^KO^ Clones

Prior to electroporation, 10.6 cells were in the log phase of growth for optimal nucleofection. To generate the RNP complex, guidelines from IDT were followed. Briefly, the custom-generated *tat* crRNA (see Appendix A for sequences) and the Alt-R^®^ CRISPR-Cas9 negative control crRNA #1 (IDT, Coralville, IA, USA, 1072544) were annealed individually to Alt-R^®^ CRISPR-Cas9 tracrRNA, ATTO™ 550 (IDT, Coralville, IA, USA, 1075927) by mixing in equal molar ratios (e.g., 5 μL each of the 100 μM stocks) to generate gRNA’s at a 50 μM concentration. Annealing was performed by heating at 95 °C for 5 min (minutes) and then cooling down to room temperature by leaving it covered on the bench for ~3 h. To prepare the RNP complex, 6 µL/30 µg/180 pmoles of TrueCut™ Cas9 Protein v2 (Thermo Fisher Scientific, Vilnius, Lithuania, A36498) and 4.5 µL/225 pmoles of the gRNA (crRNA-tracrRNA) mix were mixed and incubated at room temperature for ~20 min prior to electroporation. A 6-well plate with 1 mL of pre-warmed RPMI-10%FBS-1%P/S per well was placed in the 37 °C incubator. For electroporation, 2 × 10^6^ healthy (>95% alive) 10.6 cells were used per electroporation using the Cell Line Nucleofector^®^ Kit V (Lonza, Rockland, ME, USA, VCA-1003) on the Lonza Nucleofector 2b device. Cells were prepared by centrifugation (100× *g*, 10 min, RT), the complete aspiration of tissue culture media, and resuspension in Lonza nucleofector solution (82 µL of Nucleofector solution + 18 µL of supplement 1 per sample). For the 10.6 Tat^KO^ clone generation, 10 µL of the *tat* gRNA RNP complex (gRNA RNP) along with 1.2 µL/120 pmoles of both sense and antisense single-stranded oligonucleotide donor oligos (HDR donor oligos) (100 µM stock, see Appendix A for sequences) were added to the cell suspension, and immediately transferred to one 2 mm Lonza cuvette and electroporated using the Jurkat X-005 program. For the 10.6 Ctrl^KO^ clone generation, the negative control gRNA RNP was added to cells with no donor oligos, followed by electroporation using identical conditions to the 10.6 Tat^KO^ clone generation. Immediately after, 1 mL of 37 °C pre-warmed RPMI-10%FBS-1%P/S was added to the electroporated cell suspension, and cells were carefully collected with gel loading tips from the cuvette and placed in the pre-warmed 6-well plate for outgrowth. Two days post-electroporation, ATTO 550+ cells were sorted at the UTSW flow cytometry facility using a BD FACS Aria IIu cell sorter with a 532 nm excitation laser line and 575 nm emission filter. For all sorting experiments, cells were centrifuged (100× *g*, 5 min, RT), media was removed and then resuspended in FACS media (RPMI + 2% FBS). Standard flow cytometric parameters were used to gate for intact, single ATTO 550+ cells. These cells were expanded for ~5 days prior to treatment with either TNF-α (10.6 Tat^KO^ population) or PBS vehicle (10.6 Ctrl^KO^ population) followed by the sorting of GFP-negative cells (for both cell lines) into single cells for clonal expansion and the allowance of cells to undergo latency. Cells were expanded from single cells in 96-well plates to a confluency of ~1 × 10^6^ cells/mL in 6-well plates, which took approximately ~30 days. Clones were screened by treating them with PMA (and DMSO as the vehicle control) followed by GFP flow cytometry to identify putative 10.6 Tat^KO^ clones that induced an increased median GFP fluorescence intensity of <3-fold above basal. Putative 10.6 Tat^KO^ and 10.6 Ctrl^KO^ clones were then subjected to genomic DNA isolation, PCR, Sanger and linear amplicon sequencing, and western blot (all described below) to validate the *tat* mutation status, the fidelity of the HIV-1 genome sequence, and depletion of the Tat protein after cell stimulation.

### 2.4. Genomic DNA Isolation, PCR, Sanger Sequencing, and Amplicon Sequencing

Genomic DNA was isolated from 1 × 10^6^ cells of the clonal lines with the DNeasy Blood & Tissue Kit (Qiagen, Germantown, MD, USA 69504) following the manufacturer’s instructions. PCR (for Sanger sequencing) was performed with a Q5^®^ High-Fidelity DNA Polymerase (NEB, Ipswich, MA, USA, M0491L) and primers listed in Appendix A (3078/3079) using the following PCR program: 98 °C: 30 s (second), (98 °C: 10 s, 65 °C: 30 s, 72 °C: 60 s) for 34 cycles, 72 °C: 5 min. The ~345 bp DNA band was isolated via gel purification with the MinElute Gel Extraction Kit (QIAGEN, Germantown, MD, USA, 28606) and submitted for Sanger sequencing using primer 3079 (see Appendix A). For amplicon sequencing with Oxford nanopore, primers to generate PCR products were identified from previously published protocols for 10.6 cells [41]. Briefly, PCR was performed with the Phusion High-Fidelity PCR Master Mix and HF Buffer (Thermo Fisher Scientific, Vilnius, Lithuania, F531L) and the primers listed in Appendix A (primers 3931/3932 for the HIV-1 genome product and primers 3933/3934 for the GFP-3′ LTR product). For the ~9000 bp HIV-1 genome product, the following PCR program was followed: 98 °C: 1 min, (98 °C: 10 s, 68 °C: 20 s, and 72 °C: 4 min) for 35 cycles, and 72 °C: 1 min. For the ~2140 bp GFP-3′ LTR product, the following PCR program was followed: 98 °C: 1 min, (98 °C: 10 s, 68.8 °C: 20 s, and 72 °C: 1 min) for 35 cycles, and 72 °C: 1 min. Gel products were purified as described above (multiple PCR products were combined to increase the yield) and submitted to Plasmidsaurus for amplicon sequencing.

### 2.5. Flow Cytometry

For flow cytometry assays, 0.2 × 10^6^ cells were collected for analysis using the Stratedigm S1000 flow cytometer after treatment. The cells were collected by centrifugation (600× *g*, 5 min, RT), media was removed, and the cells were washed twice in PBS followed by additional centrifugation. Cells were resuspended in PBS and ran through the flow cytometer using channels to measure GFP and ATTO 550. FlowJo v10.8.1 was used to gate cells for intact and single cells that were positive for GFP and ATTO 550 using standard analysis techniques. Raw flow cytometry plots are also shown in the indicated figures.

### 2.6. Western Blots

Western blots were obtained through the electrophoresis of protein lysates on 12–15% SDS-PAGE and transferred on nitrocellulose membranes using the Bio-Rad Trans-Blot Turbo Transfer System, which was blocked for 1 h in 5% Milk + Tris-buffered saline-Tween-20 (TBS-T), and probed with indicated primary antibodies (Appendix A) in 5% Milk + TBS-T. The primary and secondary antibody concentration and time of incubation are indicated in Appendix A. Blots were exposed using either Clarity Western ECL (Bio-Rad, Hercules, CA, USA, 1705060) or the SuperSignal™ West Femto Maximum Sensitivity Substrate (Thermo Fisher Scientific, Rockford, IL, USA, 34095).

### 2.7. RNA Extraction and RT-qPCR

For RT-qPCR, the cells were treated as described, and RNA was extracted using the Zymo Quick-RNA MiniPrep Kit (Zymo Research, Irvine, CA, USA, R1055) following the kit’s instructions. RNA was quantified using the DeNovix DS-II FX+ Spectrophotometer, and all RNAs were diluted to the same yield and re-quantified prior to the cDNA synthesis reaction. To prepare cDNA, 2 μg of RNA was incubated with 1/200^th^ of a unit of hexanucleotide primers (MilliporeSigma, Saint Louis, MO, USA, H0268) and 1 μL of a 10 mM dNTP mix (NEB, Ipswich, MA, USA, N0447L) for 5 min at 70 °C. Next, 2 μL of a 10X M-MuLV Reverse Transcriptase buffer and 1 μL of an M-MuLV Reverse Transcriptase (NEB, Ipswich, MA, USA, M0253L) were added to each sample and incubated at 42 °C for 1 h. The reaction was inactivated at 70 °C for 10 min, and samples were diluted by adding 80 µL of H_2_O. For qPCR, 1 μL of diluted cDNA, 5 μL of the PowerUp™ SYBR™ Green Master Mix for qPCR (Thermo Fisher Scientific, Vilnius, Lithuania, A25741), 3 μL of H_2_O, and 1 μL of the 5 μM primer mix was used for each per well in a 96-well plate. The primers used for RT-qPCR analysis are listed in Appendix A. Samples were amplified for 40 cycles using the Applied Biosystems QuantStudio™ 3 Real-Time PCR System. All RT-qPCR data were analyzed using the 2^−ΔΔCt^ method where RNA for each sample was normalized as described in the figure legends, and then each target gene was normalized to two control genes (U6 and GAPDH) using the geometric mean of their expression.

### 2.8. ChIP-qPCR Assay

ChIP-qPCR was performed as previously described [42] with minor modifications. For each ChIP, cells were treated with and without PMA at a concentration of 0.9 × 10^6^ cells/mL for the described time point, and 20 × 10^6^ cells were utilized per ChIP. The vehicle condition (Time 0 PMA) was treated with DMSO for 0.5 h. After treatment, the cells were crosslinked with 0.5% methanol-free formaldehyde (Thermo Fisher Scientific, Rockford, IL, USA, 28908) by adding it directly to the cell suspension in media at room temperature for 10 min with rocking and quenching with 150 mM glycine for 5 min with rocking. After quenching, the cells were centrifuged (1000× *g*, 5 min, 4 °C), washed twice with cold PBS, flash-frozen in liquid nitrogen, and kept at −80 °C until ready to use. To perform the ChIP, cells were resuspended in 2 mL/cell pellet of a Farnham Lysis Buffer (5 mM PIPES pH = 8.0, 85 mM KCl, 0.5% NP-40, 1 mM of PMSF, 1X Protease Inhibitor (RPI, Mount Prospect, IL, USA, P50700)), counted using a hemacytometer, resuspended in 10 × 10^6^ cells/mL, nutated for 30 min at 4 °C, and then centrifuged to isolate the nuclei (1000× *g*, 5 min, 4 °C). The supernatant was removed, and nuclei resuspended in Szak’s RIPA Buffer (50 mM Tris-HCl pH = 8.0, 1% NP-40, 150 mM NaCl, 0.5% Na-Deoxycholate, 0.1% SDS, 2.5 mM EDTA pH = 8.0, 1 mM PMSF, and 1× Protease Inhibitor (RPI, Mount Prospect, IL, USA, P50700)) at a concentration of 25 × 10^6^ nuclei/mL. The chromatin was sheared using a Q800R3 Qsonica instrument (50% amplitude with 30 cycles, 30 s ON, 30 s OFF at 4 °C) to a DNA molecular weight range of 200–400 bp. After sonication, chromatin was centrifuged (21,000× *g*, 15 min, 4 °C), and the supernatant was taken as clarified chromatin. Sheared chromatin was pre-cleared by incubating with 25 μL of Szak’s RIPA equilibrated Protein G Dynabeads (Thermo Fisher Scientific, Rockford, IL, USA, 10003D) for 1 h at 4 °C. To equilibrate the beads for pre-clearing, a master mix of beads was incubated with RIPA buffer, nutated for 5 min, and placed on a magnet to remove the supernatant and the process repeated three times. To bind the antibody to beads, 100 μL of Protein G Dynabeads per sample were equilibrated with PBS + 0.05% Tween-20 and resuspended to a final volume of 250 μL per ChIP. The RPB3 antibody (see Appendix A) was then added to the 250 µL beads, incubated for 1 h at 4 °C with rotation, washed once with PBS + 0.05% Tween-20 with rotation, twice with Szak’s RIPA Buffer with rotation, and then the antibody-bound beads were then blocked in Szak’s RIPA Buffer + 5% BSA for 1 h at 4 °C with rotation. Pre-cleared sheared chromatin was then added to the blocked antibody-conjugated beads and incubated overnight at 4 °C with rotation. Beads from each sample were washed twice with 900 μL of Szak’s RIPA Buffer, Low Salt Buffer (0.1% SDS, 1% NP-40, 2 mM EDTA pH = 8.0, 20 mM Tris-HCl pH = 8.0, 150 mM NaCl), High Salt Buffer (0.1% SDS, 1% NP-40, 2 mM EDTA pH = 8.0, 20 mM Tris-HCl pH = 8.0, 500 mM NaCl), LiCl buffer (250 mM LiCl, 1% NP-40, 1% sodium deoxycholate, 1 mM EDTA pH = 8.0, 20 mM Tris-HCl pH = 8.0), and TE Buffer (10 mM Tris-HCl pH = 8.0, 1 mM EDTA pH = 8.0). After the final wash, samples were pulse-spun in a table-top centrifuge to remove the residual buffer before placing it on the magnet. Samples were then eluted from the beads in a 100 μL elution buffer (100 mM NaHCO_3_ pH = 8.0, 1% SDS) for 30 min at 65 °C while vortexing every ~10 min. Input samples (40 μL) were volumed up to 100 μL by adding 60 μL of the elution buffer. DNA was eluted by placing the beads on a magnet, transferring elutions to new tubes, and de-crosslinking for 4 h at 65 °C with a 100 μL volume of de-crosslinking buffer (500 mM NaCl, 2 mM EDTA pH = 8.0, 20 mM Tris-HCl pH = 6.8, 0.5 mg/mL Proteinase K). ChIP DNA was purified and concentrated with the Zymo ChIP DNA Clean & Concentrator (Zymo Research, San Diego, CA, USA, D5201). In total, 1 μL of each ChIP input and sample was taken for qPCR using the HIV-1 elongation primer following standard percentage input normalization.

## 3. Results

The J-Lat 10.6 model (10.6) [40], which harbors a full-length but replication-defective (Env^−^) HIV-1 (R7/3) in the second intron of human *SEC16A,* was used for the CRISPR-based approach because of its low background in basal conditions and high reactivation capacity after cell stimulation. J-Lat models are additionally attractive for studying latency reactivation given that the virus is integrated into a single locus and the genome contains a GFP reporter, enabling flow cytometric methods to approximate expression levels as a surrogate for latency reactivation.

Given that Tat-minus cells are expected to induce the virus at much lower levels than Tat-positive cells, a GFP-based selection strategy was utilized (Figure 2a) to isolate and screen for putative *tat* ATG mutants (referred to as 10.6 Tat^KO^), followed by molecular characterization to assess how Tat loss affects proviral transcription and Pol II occupancy in basal and stimulated contexts. To ensure robustness and comparability, a 10.6 Ctrl^KO^ clone was generated using a similar experimental procedure without cleavage at any locus, including HIV-1 (Figure 2b).

A crRNA targeting the *tat* ATG codon was designed using the IDT Alt-R™ CRISPR HDR Design Tool along with sense and antisense single-stranded oligonucleotide donor oligos surrounding the ATG codon with the GTC mutation (Appendix A). The crRNA and tracrRNA oligos were annealed and incubated with the Cas9 protein to form a gRNA RNP complex (gRNA RNP). Once introduced into cells via electroporation with donor oligos, the RNP complex is recruited to the *tat*-coding region of HIV-1 to generate a double-strand break, which is then repaired by the host machinery using HDR to introduce a specific mutation with the donor oligos as the template.

The tracrRNA is labeled with ATTO 550, allowing the quantitation of electroporation efficiency and cell isolation using Fluorescence-Activated Cell Sorting (FACS) to enrich those cells that received the gRNA RNP complex, thus making them more likely to be targeted by the Cas9 endonuclease (Figure 3a; raw flow plots in Appendix A). Importantly, electroporation induces morphology changes in Jurkat T cells (Appendix A); thus, sorting ATTO 550-expressing intact cells additionally enriches healthy cells to be utilized for the downstream steps in the procedure.

As a quality-control measure to test if the electroporated 10.6 Tat^KO^ population was effectively targeted, a subset of the ATTO 550+ 10.6 Ctrl^KO^ and 10.6 Tat^KO^ populations were treated with TNF-α for 16 h. This treatment induced HIV-1 expression in parental 10.6 cells, leading to a median GFP fluorescence intensity ~30–50-fold above basal, with ~60–90% of GFP+ cells. Importantly, the percentage of GFP+ cells was lower in the 10.6 Tat^KO^ population (42.1% GFP+) than in the 10.6 Ctrl^KO^ population (87.6% GFP+) (Figure 3b; raw flow plots in Appendix A), suggesting that the targeting of *tat* at least partially works. Both the ATTO 550+ 10.6 Tat^KO^ and 10.6 Ctrl^KO^ cell populations were then allowed to expand for ~5 days and treated with TNF-α (10.6 Tat^KO^) or vehicle (PBS) (10.6 Ctrl^KO^) for 16 h to activate the provirus, with the cells inducing HIV-1 at low levels (see criteria below) after 16 h of treatment in the 10.6 Tat^KO^ population defined as putative *tat* ATG mutants. The GFP negative, TNF-α treated 10.6 Tat^KO^ cells (putative *tat* ATG mutants) and GFP negative, vehicle-treated 10.6 Ctrl^KO^ cells were then individually cell sorted and grown in 96-well plates to allow for their clonal expansion and subsequent screening (Figure 3b). Importantly, the vehicle-treated 10.6 Ctrl^KO^ cells were selected for outgrowth as TNF-α strongly induces the virus in Tat-expressing cells, which may take an extended amount of time to enter latency thereby inhibiting the capability to perform downstream experiments.

To show the generality of our approach, PMA, a Protein Kinase C agonist, was used for screening purposes as it indues HIV-1 to similar levels as TNF-α in the parental 10.6 cell line; however, any latency reversal agent could be used for screening purposes. The clones were expanded and screened by treating them with PMA to identify 10.6 Ctrl^KO^ clones that were inducible (like parental 10.6) and 10.6 Tat^KO^ clones that were not inducible (induction of less than 3-fold GFP fluorescence intensity over basal as measured by flow cytometry) and potentially ATG mutants. Once putative 10.6 Tat^KO^ clones were identified, genomic DNA was isolated, and PCR was performed using primers that created a 345 bp product overlapping the *tat* ATG codon (Figure 3c) followed by Sanger sequencing to ensure that 10.6 Ctrl^KO^ cells were wild-type and to evaluate *tat* mutations in the 10.6 Tat^KO^ clones (Figure 3d). During the screening process, about 50% of the clones screened (9/17) contained *tat* ATG mutations, while the other half (8/17) contained minor deletions (ranging from 8 to 24 bp) surrounding the ATG, perhaps as a consequence of non-HR DNA repair mechanisms. An example of both clone types is shown in Figure 3d.

To validate the integrity of the genome in selected 10.6 Ctrl^KO^ and 10.6 Tat^KO^ clones, the amplicon sequencing of the entire HIV-1 provirus was performed following published protocols for the 10.6 system [41]. PCR products for both a HIV-1 genome product (5′ LTR to *env*) and a GFP-3′ LTR product revealed no additional CRISPR-mediated alterations in the clones besides the ATG to GTC conversion in the 10.6 Tat^KO^ clone (Appendix A). Additionally, the Western blots of Tat in cells treated with vehicle DMSO and PMA (to induce HIV-1 expression) showed that 10.6 Ctrl^KO^, but not 10.6 Tat^KO^ (ATG mutant) cells, expressed Tat after PMA induction, like the parental 10.6 cell line, highlighting how the genomic mutation of the *tat* ATG codon does indeed blunt Tat expression (Figure 3e). Importantly, Gag expression, which is induced with the virus, is also completely blunted in 10.6 Tat^KO^, but not 10.6 Ctrl^KO^ nor parental 10.6 cells, highlighting Tat’s contribution to HIV-1 proviral transcription upon cell stimulation (Figure 3e).

Because some clones had small Tat deletions instead of the precise mutation to inactivate the ATG codon, we interrogated their induction upon stimulation with PMA for 16 h using flow cytometry to evaluate the efficacy of using deletions to study Tat-mediated functions (Figure 4a,b; raw flow plots in Appendix A). As expected, parental 10.6 and 10.6 Ctrl^KO^ cells stimulated the provirus (~48-fold above basal), while the 10.6 Tat^KO^ clone induced only ~1.7-fold above basal (Figure 4a). The 10.6 Tat^KO^ deletion clone did not show any provirus induction upon stimulation (Figure 4a), perhaps due to the deletion eliminating splicing acceptor sites for *rev* and/or *nef* (GFP) expression (reviewed in [43]).

Flow cytometry plots in parental 10.6 cells and the three clonal lines underscore these results, which show a “shift” in the GFP levels in the 10.6 Tat^KO^ clone upon PMA treatment that is not present in the deletion (Figure 4b; Appendix A). To rule out clone-specific effects, flow cytometry and Sanger sequencing data for one additional 10.6 Tat^KO^ deletion and 10.6 Tat^KO^ clones is shown, which confirms that specific ATG mutations, but not minor deletions, lead to a modest “shift” in GFP levels upon PMA stimulation (Appendix A). Additionally, RT-qPCR data using an internal HIV-1 gene primer (HIV-1 elongation, described in Figure 5a) perfectly matched with flow cytometry data (Figure 4), whereby the 10.6 Tat^KO^ cells induced HIV-1 expression at much lower levels (less than ~100-fold) relative to 10.6 Ctrl^KO^ cells, with even lower expression levels for 10.6 Tat^KO^ deletion (Appendix A). Overall, these data suggest that some low-level host phase transcription does occur when Tat is not present, which is expected given that the host phase of HIV-1 transcription is driven by TFs induced upon cell stimulation (e.g., NF-κB, NFAT, and AP-1) prior to Tat expression [16,17,18]. However, because this behavior is lost in the 10.6 Tat^KO^ deletion clone (Figure 3d), we posit that minor deletions of the virus are not suitable for studying Tat molecular functions, emphasizing the need for the rigorous analysis of the clones generated by CRISPR-mediated genome engineering.

To test the transcriptional kinetics before and after HIV-1 induction with and without Tat, RT-qPCR was performed using a primer pair that spanned the proviral promoter (HIV-1 initiation) and internal gene primer pair in the distal *env* gene (HIV-1 elongation) in both the untreated condition (basal) as well as during a PMA time course (Figure 5a). Surprisingly, loss of Tat leads to remarkably decreased HIV-1 expression in the basal condition, suggesting that there is some low-level HIV-1 RNA synthesis in untreated cells and that Tat is important for this expression (Figure 5b). However, given that this assay is population-based, it is unclear if this expression is the average of the cells in the population or is contributed by a few stochastically expressing cells in the population. Based on studies highlighting the nature of stochastic HIV-1 expression in unstimulated cells [44] and our flow data showing that ~0.7–2% of cells often express GFP without stimulation (Appendix A), we posit that this expression arises from a small percentage of stochastic cells that express Tat, and consequently, the virus, at high levels. Regardless of the source of this expression, these data reveal previously ignored Tat expression, which could have functional implications for the control of latency reactivation.

We then investigated the HIV-1 transcriptional response with and without Tat after a PMA stimulation time course. Upon PMA treatment in 10.6 Ctrl^KO^ cells, HIV-1 RNA was detected as early as 0.5–1 h (~4–7–fold above basal), was ~70-fold above basal by 4 h, and induced almost exponentially to ~900-fold by 16 h, with similar results for both HIV-1 primer pairs (initiation and elongation) tested (Figure 5c,d). The 10.6 Tat^KO^ cells induced HIV-1 at the early time points (~8–10 fold above the basal), similar to 10.6 Ctrl^KO^ cells, but then stagnated (HIV-1 initiation) or decreased close to basal levels (HIV-1 elongation) over time (Figure 5c,d). These data suggest that by ~4 h, the host phase had already begun winding down and that Tat was responsible for driving HIV-1 transcription to the high levels seen at 4 and 16 h in 10.6 Ctrl^KO^ cells, suggesting that the transition point between host and viral phases lies somewhere between the 1 h and 4 h time points in this cell system and with the specific conditions used. Importantly, similar results were obtained for two additional HIV-1 genes, *rev* and *vpr* (Appendix A). The above RT-qPCR data were normalized to respective 0 time points (e.g., each 10.6 Ctrl^KO^ PMA time point to 10.6 Ctrl^KO^ 0 h PMA; each 10.6 Tat^KO^ PMA time point to 10.6 Tat^KO^ 0 h PMA), which ignored the decreases in transcription observed without Tat in the basal phase (Figure 5b). Strikingly, data normalization to the 10.6 Ctrl^KO^ 0 time point highlights the drastic decrease in total HIV-1 mRNA expression when Tat was absent, suggesting that low levels of Tat in basal may be key to driving robust levels of HIV-1 activation (Appendix A). These insights could only be gleaned using a system where Tat could be marked as “null” prior to the cell stimulation, thus reinforcing the idea that our approach enabled a distinction between the host and viral phases of the HIV-1 transcriptional program.

Importantly, the transition point defined using PMA may not be the same for every ligand that induces latent HIV-1 reactivation. To test this idea, a similar RT-qPCR time course experiment was performed using TNF-α, a pro-inflammatory cytokine that activated latent HIV-1 reactivation to similar levels as PMA. While the overall magnitude of induction was lower for TNF-α in this cell system, the time we observed the host phase winding down and the viral phase beginning (the transition point for TNF-α–induced latent HIV-1 reactivation) was also between the 1 and 4 h time points (Figure 5e,f). While both TNF-α and PMA induced latent HIV-1 reactivation in the 10.6 model, which led to similar transition points, we emphasize that this could largely change with the use of different ligands, doses, and/or cell models of latency. Future studies will interrogate these ideas.

To obtain mechanistic insights between the host-viral phase transition beyond measuring RNA expression, ChIP assays were performed using an antibody towards Pol II (RPB3 subunit) upon the PMA stimulation time course in both 10.6 Ctrl^KO^ and 10.6 Tat^KO^ clones. Following ChIP, qPCR was performed for the internal HIV-1 genome primer (HIV-1 elongation, Figure 6) to probe Pol II occupancy as it transcribes HIV-1 during cell stimulation. In 10.6 Ctrl^KO^ cells, Pol II levels increased in the proviral genome as early as 0.5 h of PMA stimulation (~39-fold above basal), which is in line with increased transcription at this time point (Figure 5d). By 1 h PMA, Pol II levels decreased to ~9-fold above the basal but increased back to ~38-fold and ~65-fold above basal after 4 h and 16 h of PMA treatment, respectively (Figure 6). In 10.6 Tat^KO^ cells, Pol II levels only increased at the 0.5 h PMA time point (~6.2-fold above basal), after which Pol II levels reached basal levels (Figure 6). Together, these data suggest that the host phase begins winding down after ~1 h (given the decrease in the Pol II signal between 0.5 and 1 h PMA in 10.6 Ctrl^KO^ cells) and that the viral phase kicks off after the 1 h time point, which is in perfect agreement with the transition point defined by RT-qPCR data in Figure 5. In 10.6 Tat^KO^ cells, Pol II occupancy falls to basal levels after 1 h of PMA, supporting the notion that the host phase has been shut off by this point and highlighting the utility of the 10.6 Ctrl^KO^/Tat^KO^ pair of cell lines in enabling a clear distinction between the host and viral phases of the HIV-1 transcriptional circuit for future mechanistic studies.

## 4. Discussion

Here, we implemented a proviral genome engineering CRISPR approach to eliminate Tat expression via the introduction of mutations in the *tat* ATG codon. By profiling HIV-1 expression and Pol II occupancy in a genetically matched pair of T cell clones harboring HIV-1 at the same integration site with and without Tat, we defined the transition point between the host and viral phases in the complex HIV-1 transcriptional program (Figure 1, Figure 5, and Figure 6). The advantage of this approach to the field is that it can be exploited to determine whether select factors contribute to the activation of the integrated provirus during the host and/or viral phases of the transcriptional program. Understanding this knowledge could help guide the identification of factors for their therapeutic manipulation for either reactivation or durable silencing approaches to achieve long-awaited drug-free HIV-1 remission [45,46]. This system can also help validate that select LRAs and LPAs truly operate through the expected host factors to promote latency reversal (by igniting the host phase) or durable silencing (by restraining the host phase or Tat to prevent the feedback loop during the viral phase), respectively. Additionally, our method has technical advantages as the implementation of this protocol is relatively fast, as a cell line can be made in less than 1.5 months. Also, while we use flow cytometry to enrich electroporated cells and TNF-α treatment to enrich putative *tat* ATG mutants, the efficiency is high enough (Figure 3b) such that the protocol could be conducted without cell-sorting events.

By profiling HIV-1 expression and Pol II occupancy in the genetically matched pair of T cell clones, we made a number of salient observations. First, Tat inactivation largely decreased HIV-1 expression in the basal condition, which is perhaps attributed to the elimination of the small pool of stochastically activated Tat-positive cells [44]. Second, Tat inactivation eliminates the onset of the viral phase and positive feedback loop for both Pol II occupancy and RNA expression, which is consistent with a large body of work [6,7,8,9,10,11]. Notably, our quantitative data revealed that Tat contributes exorbitantly to the viral phase as RNA levels increased by ~3 orders of magnitude in 10.6 Ctrl^KO^ but not 10.6 Tat^KO^ cells at late (16 h) time points. Third, Tat inactivation helps define the duration of the host phase, which is rapidly turned off due to the inability to transition to the viral phase without Tat (Figure 1). In the absence of Tat, the induction of the host phase upon cell stimulation does occur as expected with low levels of HIV-1 induction, likely due to the recruitment of host TFs such as NF-κB that are independent of Tat expression and are needed to induce the initial rounds of transcription leading to Tat synthesis and subsequent robust activation [47]. Notably, in this context, HIV-1 behaves much like an inducible human gene whose expression is reversible [48]. Fourth, the CRISPR approach yields both precise *tat* ATG mutations and small deletions surrounding the ATG codon, perhaps because of non-HR-mediated DNA repair mechanisms. We noticed striking functional differences between the *tat* ATG mutation and small deletion. Future work beyond the scope of this functional method’s description would be required to investigate what mechanisms (e.g., splicing defects dampening *rev* expression) drive these differences in HIV-1 gene expression.

Notably, future studies will leverage this system along high-resolution genomic approaches to interrogate how host factors and Tat modulate Pol II function prior to and after reservoir cell stimulation, including promoter recruitment, pausing and pause release, elongation rate and processivity, termination, and potential recycling for new rounds of initiation. This system will additionally allow the dissection of how TFs in the host phase and Tat in the viral phase regulate non-Pol II-mediated transcriptional processes, including transcription factor binding, chromatin accessibility, chromatin compaction, and phase separation, among others. Importantly, these molecular events can be investigated in the context of host factor depletion using short-term RNAi or advanced acute depletion methods (such as dTAG [49]) to functionally link factors to the precise regulatory phases of the HIV-1 transcriptional program.

Our CRISPR approach has many additional applications. First, it can be applied to eliminate the expression of other HIV-1 proteins (e.g., *vpu*, *vpr*) or genomic elements (e.g., the TAR element or cis-binding motifs for sequence-specific TFs at the 5′-LTR) to study how individual viral proteins or protein-binding sites regulate discrete steps of the HIV-1 transcriptional program, as well as other functions during the viral life cycle. Second, it can be applied to study a broad range of viruses to delineate how their virally encoded TFs [50] differentially regulate transcription and/or other life cycle processes.

Finally, we anticipate this system of a genetically matched pair of T cell clones will allow for the discovery and characterization of host drug targets that either function or not alongside Tat to activate HIV-1. The proper activation of the host phase is critical for full HIV-1 transcription and reactivation from latency. Thus, the identification of factors that operate during the host phase, prior to Tat synthesis, are important therapeutic targets to silence HIV-1 for the long-awaited, durable, drug-free remission to prevent Tat action [45,46]. Our cell-based system enables the distinction between multiple phases of the HIV-1 transcription cycle to address these critical, therapeutically driven questions.

## Figures and Tables

**Figure 1 viruses-16-00287-f001:**
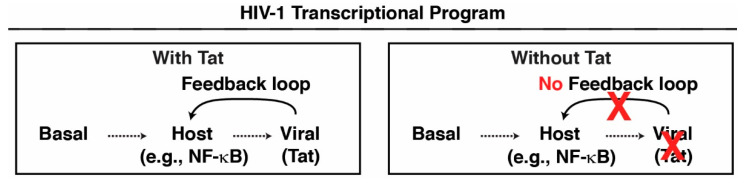
The HIV-1 transcriptional program consists of three phases: basal, host, and viral. Cellular TFs facilitate activation of the host phase to synthesize Tat and promote the viral phase and positive feedback loop. Without Tat, the viral phase is never turned on, thereby preventing the feedback loop and sustained HIV-1 transcription activation.

**Figure 2 viruses-16-00287-f002:**
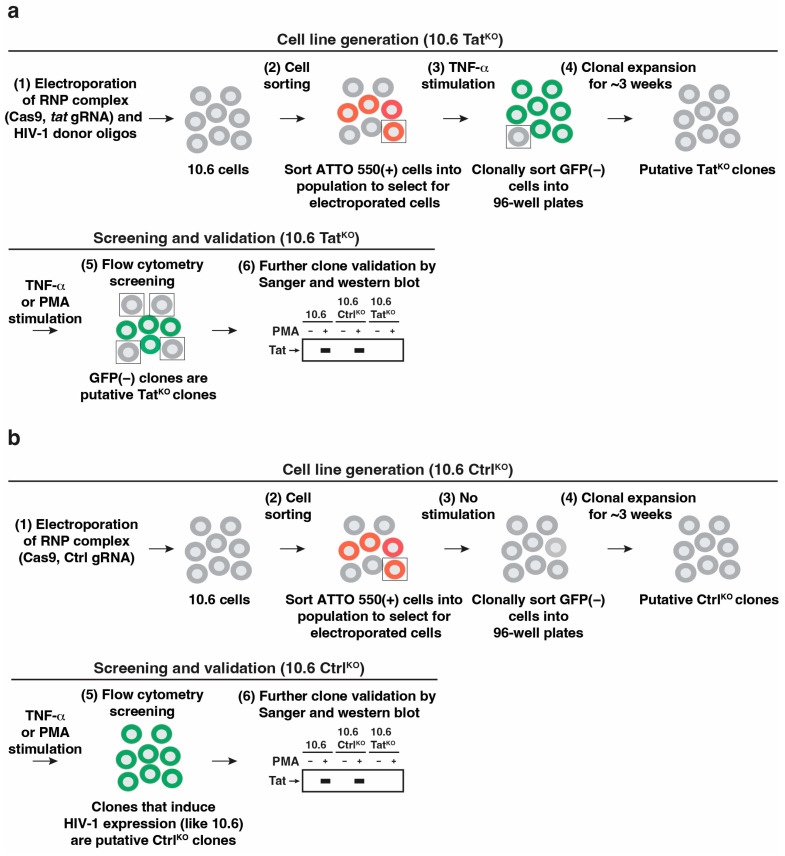
A flow cytometry-based method for the generation of Tat-null 10.6 clones. (**a**) Scheme for the generation of 10.6 Tat^KO^ cells. (**b**) Scheme for the generation of 10.6 Ctrl^KO^ cells.

**Figure 3 viruses-16-00287-f003:**
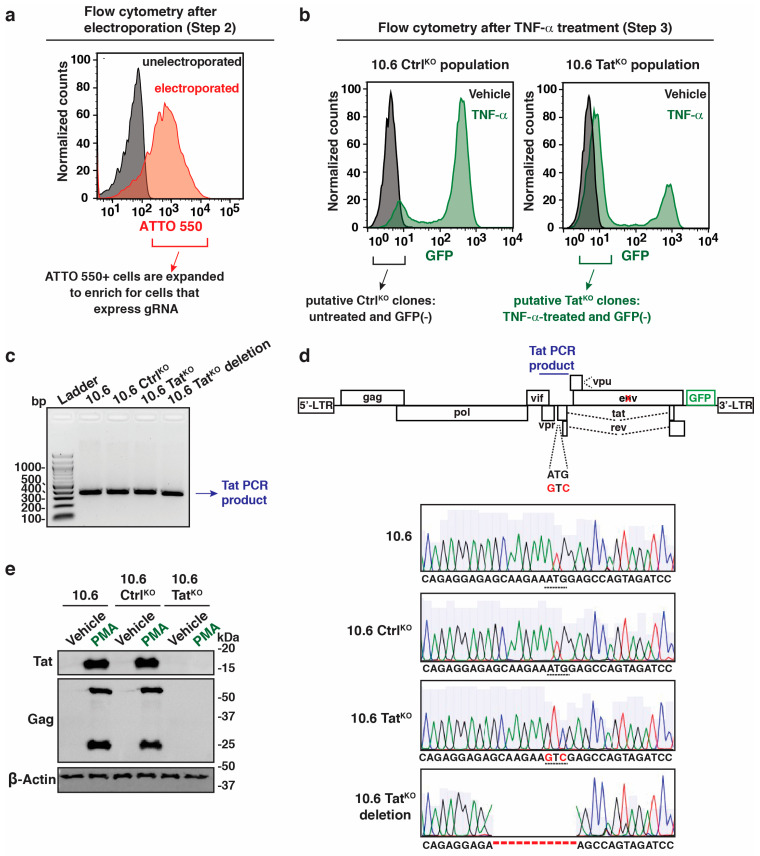
The 10.6 Tat^KO^ cells can be generated using a CRISPR-based approach. (**a**) Flow cytometry histogram showing ATTO 550 levels upon electroporation with the gRNA RNP complex. (**b**) Flow cytometry histogram(s) of 10.6 Ctrl^KO^ and 10.6 Tat^KO^ ATTO 550+ cell populations treated with vehicle or TNF-α for 16 h. (**c**) Agarose gel showing the 345 bp PCR product of 10.6, 10.6 Ctrl^KO^, 10.6 Tat^KO^, and 10.6 Tat^KO^ deletion clones. (**d**) The HIV-1 proviral genome scheme and Sanger sequencing results for the 4 cell lines in panel (**c**). The cross (X) indicates a mutation that prevents *env* expression. (**e**) Western blot of Tat, Gag, and β-Actin in 3 cell lines (10.6, 10.6 Ctrl^KO^, and 10.6 Tat^KO^) treated with vehicle or PMA for 16 h.

**Figure 4 viruses-16-00287-f004:**
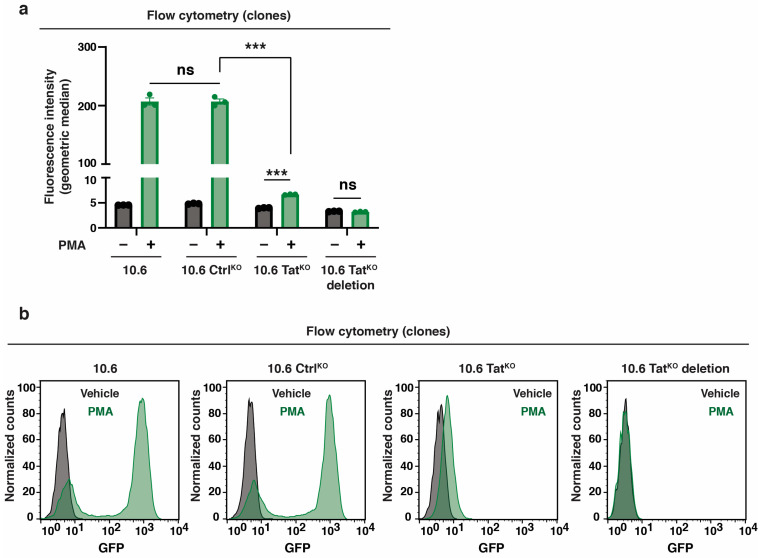
Precise *tat* genomic mutations and large deletions lead to differences in HIV-1 induction upon cell stimulation. (**a**) Flow cytometry quantitation of GFP fluorescence intensity (geometric median) in 4 cell lines (10.6, 10.6 Ctrl^KO^, 10.6 Tat^KO^, and 10.6 Tat^KO^ deletion) −/+ 16 h in PMA. Unpaired Student’s t-test, *n* = 3, −/+ SEM, ns = non-significant, *** *p* < 0.001. (**b**) Flow cytometry histograms of the data in panel (**a**).

**Figure 5 viruses-16-00287-f005:**
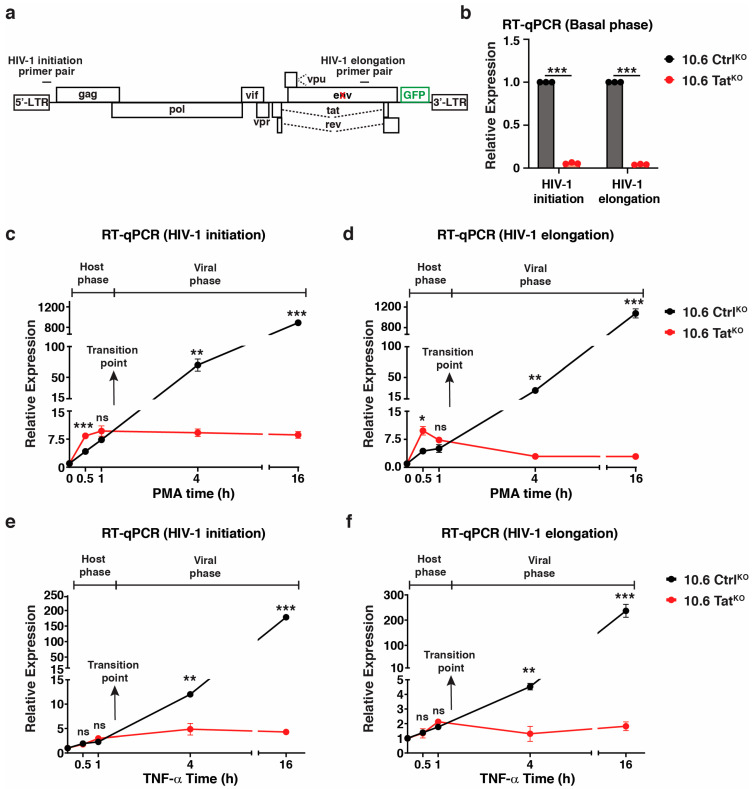
Tat sustains enhanced HIV-1 expression upon cell stimulation. (**a**) HIV-1 genome organization highlighting the genomic location of HIV-1 primers used in RT-qPCR analysis. The cross (X) indicates a mutation that prevents *env* expression. (**b**) RT-qPCR in 10.6 Ctrl^KO^ and 10.6 Tat^KO^ cells prior to cell stimulation. Unpaired Student’s t-test, *n* = 3, −/+ SEM, *** *p* < 0.001. (**c**,**d**) RT-qPCR for HIV-1 initiation (**c**) and HIV-1 elongation (**d**) in 10.6 Ctrl^KO^ and 10.6 Tat^KO^ cells across the PMA time course. All data presented in these plots are normalized to the respective vehicle control for that cell line. Statistics are compared between 10.6 Ctrl^KO^ and 10.6 Tat^KO^ for each time point using an unpaired Student’s t-test, *n* = 3, −/+ SEM, ns = non-significant, * *p* < 0.05, ** *p* < 0.01, *** *p* < 0.001. (**e**,**f**) RT-qPCR for HIV-1 initiation (**e**) and HIV-1 elongation (**f**) in 10.6 Ctrl^KO^ and 10.6 Tat^KO^ clones across the TNF-α time course. All data presented in these plots are normalized to the respective vehicle control for that cell line. Statistics are compared between 10.6 Ctrl^KO^ and 10.6 Tat^KO^ for each time point using an unpaired Student’s t-test, *n* = 3, −/+ SEM, ns = non-significant, ** *p* < 0.01, *** *p* < 0.001.

**Figure 6 viruses-16-00287-f006:**
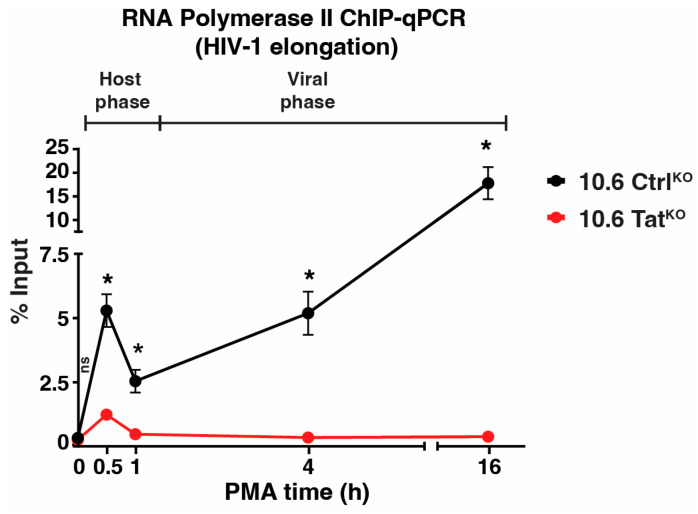
Tat sustains enhanced Pol II levels at the HIV-1 proviral genome upon cell stimulation. Pol II ChIP-qPCR using the HIV-1 elongation primer in the 10.6 Ctrl^KO^ and 10.6 Tat^KO^ clones across the PMA time course. Data presented are normalized to the percentage input. Statistics are compared between both clones for each time point using unpaired Student’s t-test, *n* = 2, −/+ SEM, ns = non-significant, * *p* < 0.05.

## Data Availability

The primary uncropped images and gel blots can be found online at MDPI.

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
