# Peer review of "HIV-1 Proviral Genome Engineering with CRISPR-Cas9 for Mechanistic Studies"

_viruses, 2024, doi:10.3390/v16020287_

Round 1
Reviewer 1 Report
Comments and Suggestions for Authors
The authors used CRISPR-CAS9 to induce a specific mutation at the Tat start codon to discern the contribution of host factors as well as Tat during HIV transcriptional regulation. They obtained the expected clones in which the Tat start codon was mutated and a few clones with a short deletion. This study is simplistic, it does not include a lot of mechanistic insight as was proposed to be the goal of making such clones (meaning no nucleosome positioning, chromatin accessibility, etc), however it is interesting, and if these clones are made available they are a useful tool to study Tat function on HIV. Here are some minor suggestions:
1. In Fig 3d, the authors identify the mutation by PCR of a short HIV gDNA spanning Tat, which is a good approach but not enough. It is important to demonstrate, on the selected clones, the full sequencing of the entire HIV genome, to confirm the mutations are at the Tat start codon only.
2. I suggest the authors include the results of mRNA levels if Fig 4a.
3. In Fig 5b, loss of Tat lead to remarkably decreased HIV-1 expression in the basal conditions, suggesting that there is some low-level HIV-1 RNA synthesis in untreated cells and that Tat is important for this expression” (from text lines 296-298). I agree with the authors’ opinion that “this expression arises from a small percentage of stochastic cells that express Tat, and consequently the virus, at high levels” (lines 303-304). This work could be enhanced with the use of additional cell models with different HIV basal levels and integration sites. In sum, similar work using different HIV latency models would be useful.
4. Figs 5c-d, the authors concluded that the transition point between host and viral phase lies somewhere between the 1hr and 4hr point in this system (lines 324-325). However, the time point might be dependent on the type of LRA used, at it may results in different speeds of mRNA accumulation. To solidify such statements I suggest including experiments with different LRA such as TNF-a, TSA etc. since HIV transcription is driven by various host factors.
5. Fig 3d, and lines 253-254, “about 50% of the putative TatKO clones contained deletions.” How many clones of deletions were identified? How is the HIV expressed in these clones? It’s true that the minor deletions of the virus are not suitable for studying Tat functions. However, it is still interesting to study if Rev and Vpr (both at mRNA and protein level) are affected in these clones. If this data was generated, a discussion could be included.
6. It is well known that the anti-Tat antibody (ab43014) has a very strong unspecific band between 10-15KDa. So please explain is this is the case Fig 3e and/or suggest using another antibody.
Author Response
We have addressed all requests as indicated in the attachment.

Reviewer 2 Report
Comments and Suggestions for Authors
The manuscript by Hyder and co-workers entitled "HIV-1 proviral genome engineering with CRISPR-Cas9 for mechanistic studies" is focused on the generation, by CRISPR approach, of genetically matched pair of T cell clones harboring HIV-1 at the same integration site, with and without Tat. The topic of the work is interesting, as a better understanding of the complex virus/intrinsic host factors interplay involved in viral latency coud contribute to the development of safe and more effective and targeted interventions to eliminate HIV. The manuscript is well written, the study is well conducted, the experimental parts appear overall robust and the conclusions supported.
The manuscript may be suitable for publication. However, according to this Referee, few points need to be addressed before been published:
1. It would be useful to give details of the J-Lat 10.6 cells in the Methods section.
2. It should be better to specify how many 10.6 TatKO and 10.6 CtrlKO clones were obtained. In the Results section, it is mentioned: “Additionally, about ~50% of the putative TatKO clones contained deletions (ranging from 8-24 bp) surrounding the ATG (one example, is shown in Figure 3d)”…how many clones with the deletions ? Only one clone “for each type” has been reported: do the other ones behave in a similar manner ? If this is the case, please specify it.
3. Considering that the T cell clones have been properly characterized and are proposed to be adopted as a system to distinguish between the host and viral phases of the HIV-1 transcription program, the inclusion of an example of such an application in this regard might improve the quality of the manuscript.
Minor points:
1. The page numbers are not always reported in a similar manner: as examples, see Ref #4, 12, 22, 23, 36, 47; in addition, for Ref #38, 43, 44, is there a page number to be indicated ?
2. In supplementary Table 1, Number 3078: HIV-1_VIF_For, in the opinion of this Referee should be HIV-1_TAT_For
In the opinion of this Referee, this manuscript by Hyder et al. is scientifically valid and could be of interest for Readers of Viruses. Nevertheless, according to this Referee, few points need to be addressed and/or clarified in order to render the manuscript suitable for publication.
Author Response
We have addressed all the requests as indicated in the attachment.

Round 2
Reviewer 1 Report
Comments and Suggestions for Authors
The authors did a very good job addressing the concerns and revise the manuscript.